# Novel Galactopyranoside Esters: Synthesis, Mechanism, In Vitro Antimicrobial Evaluation and Molecular Docking Studies

**DOI:** 10.3390/molecules27134125

**Published:** 2022-06-27

**Authors:** Priyanka Matin, Umme Hanee, Muhammad Shaiful Alam, Jae Eon Jeong, Mohammed Mahbubul Matin, Md. Rezaur Rahman, Shafi Mahmud, Mohammed Merae Alshahrani, Bonglee Kim

**Affiliations:** 1Bioorganic and Medicinal Chemistry Laboratory, Department of Chemistry, Faculty of Science, University of Chittagong, Chittagong 4331, Bangladesh; priyankapeu57@gmail.com (P.M.); ummehanee19@gmail.com (U.H.); 2Department of Pharmacy, University of Science and Technology Chittagong, Chittagong 4202, Bangladesh; shaifulalam.mmddl@gmail.com; 3Department of Pathology, College of Korean Medicine, Kyung Hee University, Seoul 02447, Korea; jaeeon0511@khu.ac.kr; 4Department of Chemical Engineering and Energy Sustainability, Faculty of Engineering, Universiti Malaysia Sarawak, Kota Samarahan 94300, Malaysia; rmrezaur@unimas.my; 5Division of Genome Sciences and Cancer, The John Curtin School of Medical Research, The Shine-Dalgarno Centre for RNA Innovation, The Australian National University, Canberra, ACT 2601, Australia; shafi.mahmud@anu.edu.au; 6Department of Clinical Laboratory Sciences, Faculty of Applied Medical Sciences, Najran University, Najran 61441, Saudi Arabia; mmalshahrani@nu.edu.sa

**Keywords:** ADMET studies, antifungal agents, dynamics simulation, methyl α-D-galactopyranoside esters, molecular docking, one-step acylation, SARS-CoV-2

## Abstract

One-step direct unimolar valeroylation of methyl α-D-galactopyranoside (MDG) mainly furnished the corresponding 6-*O*-valeroate. However, DMAP catalyzed a similar reaction that produced 2,6-di-*O*-valeroate and 6-*O*-valeroate, with the reactivity sequence as 6-OH > 2-OH > 3-OH,4-OH. To obtain novel antimicrobial agents, 6-*O*- and 2,6-di-*O*-valeroate were converted into several 2,3,4-tri-*O*- and 3,4-di-*O*-acyl esters, respectively, with other acylating agents in good yields. The PASS activity spectra along with in vitro antimicrobial evaluation clearly indicated that these MDG esters had better antifungal activities than antibacterial agents. To rationalize higher antifungal potentiality, molecular docking was conducted with sterol 14α-demethylase (PDB ID: 4UYL, *Aspergillus fumigatus*), which clearly supported the in vitro antifungal results. In particular, MDG ester **7**–**12** showed higher binding energy than the antifungal drug, fluconazole. Additionally, these compounds were found to have more promising binding energy with the SARS-CoV-2 main protease (6LU7) than tetracycline, fluconazole, and native inhibitor N3. Detailed investigation of Ki values, absorption, distribution, metabolism, excretion, and toxicity (ADMET), and the drug-likeness profile indicated that most of these compounds satisfy the drug-likeness evaluation, bioavailability, and safety tests, and hence, these synthetic novel MDG esters could be new antifungal and antiviral drugs.

## 1. Introduction

Carbohydrates, especially natural analogs, have high polarity and lability of the *O*-glycosidic bond, leading to inappropriate transport to the pharmacological site of action [1,2], and thus, restricting their application as potential drugs. Ultimately, the search for more stable and less hydrophilic analogs than common carbohydrates is essential [3,4,5,6]. These drawbacks are generally overcome by acyl sugars or sugar esters (SEs), which are composed of one or more fatty acids as hydrophobic moieties with a carbohydrate skeleton. Nowadays, the booming areas of usage of carbohydrate materials are the biomedical or medical fields, where SEs are significantly used as a versatile and valuable intermediate for the modern target-orientated synthesis of many bioactive natural products [4,5,6,7,8,9]. SEs are non-toxic, biodegradable, amphiphilic, nonirritating, environment friendly, and have a longer shelf life [10]. Due to their good stabilizing and conditioning properties, SEs have increased inclusion in daily commodities including as low caloric sweeteners, flavorings, and biosurfactants/emulsifiers in foods, and food additives, detergents, and in pharmaceutical, biomedical, cosmetic, and oral-care products [11,12,13]. More therapeutic potentiality such as protective effects against neurodegeneration [14], anti-breast cancer effects, and antioxidant properties [15] revealed their significant biomedical applications. Various carbohydrate-binding agents have been shown to possess many binding affinity sites with many carbohydrate complexes with the spike glycoprotein of severe acute respiratory syndrome coronavirus 2 (SARS-CoV-2) [16].

Among the natural and synthetic carbohydrate derivatives, galactopyranoside-based esters draw special attention [17] and are investigated for biological activities including α-galactosidase inhibition [18]. For example, umbelliferone β-D-galactopyranoside (UFG) isolated from the bark of *Aegle marmelos* (L.) possesses antidiabetic, antioxidant, and antihyperlipidemic effects [19]. Whereas, 6-*O*-acyl-1,2:3,4-di-*O*-isopropylidene-α-D-galactopyranoside (**1**, Figure 1) was found to contain weak antibacterial activities [20,21]. The addition of the acyl group(s) at the C-6 position of galactopyranoside **2** slightly distorted the ^4^*C*_1_ chair conformation [22], and the presence of two isopropylidene rings, as in **1,** strongly distorted the pyranose ring from the regular ^4^*C*_1_ conformation, which leads to poor antibacterial functionality [21]. However, a non-protected form such as *N*-acyl derivatives of diosgenyl 2-amino-2-deoxy-β-D-galactopyranoside indicated activities against bacteria and substantially against *Candida*-*type* fungi [23]. Furthermore, the 5-aminosalicylic acid (5-ASA) esters of galactopyranose were found to be highly active in the treatment of inflammatory bowel disease (IBD) and showed notable antimicrobial efficacy with minor side effects [24]. Thus, galactopyranose esters could be an alternative choice to azole drugs with many side effects.

Several synthetic methods were developed for the esterification of sugars including selective and regioselective esterification [25,26,27,28,29,30]. The existence of several 2° hydroxyl groups of almost similar reactivity greatly affected the selective and regioselective esterification of carbohydrates [31]. The huge variation in the structures of carbohydrate molecules makes it difficult to obtain selectivity. Hence, the direct synthetic method is favored to reduce the number of steps and increase the product yield(s) [32,33,34]. Some of the SEs were found to be active against multidrug-resistant (MDR) pathogens [35], especially fungi [14,30,32], and hence provided a promising prospect for non-azole-type sugar-based antimicrobial agents.

However, variable results were reported for SEs against different bacterial, fungal, and viral species [36], despite their wide range of applications. Suitable explanations of their applications by the structure–activity relationship (SAR) with molecular docking and pharmacokinetic studies are rarely reported. Hence, it is very reasonable to study new synthetic techniques with improved yield, SAR, mode of action with positional, and chain length effects of the ester group(s), supported by molecular docking and pharmacokinetic calculation of the SEs. Considering these facts, the search for non-azole-type drugs against MDR pathogens, and the current SARS-CoV-2 pandemic situation, galactopyranoside **3** was chosen as the glycon component and incorporated into several ester groups as the pharmacophore (aglycon/lipophilic component). Thereafter the novel esters were used for the abovementioned studies mainly to establish insights into the antifungal functionality of the CYP51 inhibitor to overcome MDR microbial strains and to obtain the SARS-CoV-2 main protease inhibitor.

## 2. Results and Discussion

### 2.1. Selective 6-O-Valeroylation of MDG

One-step valeroylation of MDG (**3**) was conducted by the treatment of **3** with unimolar valeroyl chloride in pyridine at low temperature (0 °C) for 12 h, which upon chromatographic purification produced a solid, mp 95–96 °C (Figure 1).

Its FT-IR spectrum showed characteristic bands at 3200–3450 (br OH), 1739 (CO), and 1048 cm^−1^ (pyranose ring; Appendix A), and hence indicated the attachment of only one valeroyl group in the compound. A two-proton triplet at δ 2.35, two two-proton multiplets at δ 1.58–1.69, and 1.27–1.43, and a three-proton triplet at δ 0.92, totaling an additional nine protons in its ^1^H NMR spectrum, further confirmed the attachment of one valeroyloxy group in the molecule. Moreover, H-6 (δ 4.36) and H-6′ (δ 4.27) resonated at considerable higher frequencies as compared to the precursor **3** (H-6 at δ ~3.69, and H-6′ at δ ~3.28) [37,38] and clearly demonstrated the incorporation of the valeroyloxy group at the C-6 position (Appendix A). This fact was further supported by analyzing its ^13^C NMR spectrum, which exhibited the additional characteristic signals at δ 173.9 (*C*O), 33.9 (*C*H_2_), 27.0 (*C*H_2_), 22.2 (*C*H_2_), and 13.6 (*C*H_3_). Thus, from FT-IR, ^1^H, and ^13^C NMR spectra (Appendix A), and C/H analyses, the compound was unambiguously assigned as methyl 6-*O*-valeroyl-α-D-galactopyranoside (**4**).

It should be noted that the moderate yield in the above reaction (Figure 1) was due to the formation of a small amount (10–15%) of the complex products (with higher *R*_f_) and some of the starting compounds were recovered at low temperature. With the increase in the reaction temperature, several products were formed as the acylating agent is not bulky.

### 2.2. DMAP-Catalyzed Valeroylation, Reactivity Order of MDG, and Probable Mechanism

A similar one-step unimolar valeroylation of MDG (**3**) in the presence of 4-(dimethylamino)pyridine (DMAP) in pyridine at low temperature (0 °C) for 16 h gave two faster moving products with *R*_f_ = 0.59 and 0.40 (chloroform/methanol = 5/1; Figure 2). The first elution with chloroform/methanol (20:1) provided a faster-moving component (*R*_f_ = 0.59) in a 56% as a colorless solid, mp 58–59 °C.

FT-IR bands at 3250–3570 (br OH), 1736, 1735 (CO), and 1044 cm^−1^ (pyranose ring) indicated the attachment of two valeroyl groups in the MDG molecule. This fact was further supported by analyzing its ^1^H NMR, where an additional eighteen protons were observed in the aliphatic region, corresponding to two valeroyl groups. The appearance of related carbon signals at δ 174.4, 173.9 (2×*C*O), 34.0, 33.9 (2×*C*H_2_), 27.0, 26.9 (2×*C*H_2_), 22.2, 22.1 (2×*C*H_2_), and 13.6(2) (2×*C*H_3_), in its ^13^C NMR spectrum, confirmed the attachment of two valeroyloxy groups. The reasonable shift of H-2 to the higher frequency at δ 5.03 from its usual value (~δ 3.88), H-6 to δ 4.45, and H-6′ to δ 4.24, as compared to galactopyranoside **3** [14,26], clearly suggested the attachment of valeroyl groups at the C-2 and C-6 positions (Appendix A), which fully corroborated the COSY, HSQC, and HMBC experiments (Appendix A). On the basis of its FT-IR ^1^H and ^13^H NMR spectra, the structure of the compound was established as methyl 2,6-di-*O*-valeroyl-α-D-galactopyranoside (**5**).

Additional elution of the mixture with chloroform/methanol (12:1) provided the slower-moving component (*R*_f_ = 0.40) as a solid (29%; Figure 2). Physical and spectroscopic data of this compound were found to be completely identical with **4,** as prepared earlier by the direct method. It should be noted that tin-mediated acylation generally produces site-selective C-3 OH acylation for MDG **3** [38], thus, DMAP-catalyzed valeroylation showed an advantage for site-selective acylation at C-2 OH along with primary OH at the C-6 position. Overall, the formation of 6-*O*-valeroate **4** and 2,6-di-*O*-valeroate **5** demonstrated that the reactivity order of the OH groups in MDG is 6-OH > 2-OH > 3-OH,4-OH for DMAP-catalyzed valeroylation.

To explain the DMAP-catalyzed formation of 2,6-di-*O*-valeroate **5,** we visualized its probable mechanism (Figure 2) in accordance with the previous observation [39] that initially 6-*O*-valeroate **4** is formed. The *N*-acyl(4-dimethylamino)pyridinium ion then selectively attacks the less hindered equatorial C-2 OH position of **4** to form a complex **4a**. Finally, the abstraction of a proton from **4a** by pyridine facilitates the removal of DMAP to produce compound **5**.

### 2.3. Synthesis of 2,3,4-Tri-O-acyl Esters from 6-O-Valeroate ***4***

Having 6-*O*-valeroate **4** in hand, we synthesized three 2,3,4-tri-*O*-acyl esters, **6**–**8,** employing acetic anhydride, hexanoyl chloride, and octanoyl chloride to introduce saturated ester chains (2C–8C) in the molecule. Firstly, triol **4,** on reaction with acetic anhydride in pyridine in the presence of DMAP for 12 h, afforded a faster-moving single product compound as a syrup (92% yield; Figure 3).

The absence of the OH group(s) frequency and the presence of four carbonyl stretching peaks at 1755, 1754, 1745, and 1743 cm^−1^ in the FT-IR spectrum (Appendix A) indicated the tri-*O*-acetylation of the molecule. This was supported by the appearance of an additional three three-proton singlets at δ 2.15, 2.10, and 1.99 in its ^1^H NMR spectrum (Appendix A). Finally, the appearance of related carbon signals at δ 170.4, 170.1, 169.9 (CH_3_*C*O), 20.8, and 20.6(2) (*C*H_3_CO) confirmed the attachment of three acetyl groups in the molecule. At this stage, the position of the attachment of the acetyl groups was established by (i) the considerable shift of H-2 (δ 5.18), H-3 (δ 5.36), and H-4 (δ 5.45) protons to higher frequencies as compared to precursor **4**, and (ii) the HMBC experiments (Figure 3 and Appendix A). Thus, the structure of the syrup was established as methyl 2,3,4-tri-*O*-acetyl-6-*O*-valeroyl-α-D-galactopyranoside (**6**) (Appendix A).

Similarly, the separate treatment of **4** with 3.3 molar eq. of hexanoyl chloride and octanoyl chloride gave corresponding tri-*O*-hexanoate **7** (Appendix A) and tri-*O*-octanoate **8** (Appendix A) respectively, in very good yields (Figure 3).

### 2.4. Synthesis of 3,4-Di-O-acyl Esters from 2,6-di-O-Valeroate ***5***

We exploited the free C-3 and C-4 OH positions of compound **5** for further esterification with different saturated chain lengths (2C to 10C). Initially, the dimolar reaction of acetic anhydride with **5** afforded a syrup in a good yield (Figure 4).

The absence of OH stretchings and the presence of carbonyl characteristic peaks at 1737, 1733, 1729, and 1721 cm^−1^ indicated complete acetylation of the molecule (Appendix A). The proton NMR (Appendix A) had two additional three-proton singlets at δ 2.15 and 1.98, and carbon NMR (Appendix A) indicated two additional carbonyl (*C*O) signals at δ 170.2 and 169.9, and two methyl carbon signals at δ 20.6 and 20.5, which confirmed the attachment of two acetyl groups in this compound. Furthermore, a higher frequency shift of H-2 (δ 5.16) and H-4 (δ 5.45) indicated the attachment of the acetyl group at the C-3 and C-4 positions, respectively, fully corroborating the COSY, HSQC, and HMBC experiments (Appendix A). Therefore, the structure of the compound was assigned as methyl 3,4-di-*O*-acetyl-2,6-di-*O*-valeroyl-α-D-galactopyranoside (**9**).

With these successes, we employed hexanoyl chloride, octanoyl chloride, and decanoyl chloride for 3,4-position acylation and obtained MDG ester **10**, **11,** and **12**, respectively, in good yields (Figure 4), which were in complete accord with their spectral (Appendix A) and elemental analyses.

### 2.5. Prediction of Antimicrobial Activities of ***3**–**12***

Initially, we predicted the antimicrobial potentialities of MDG-based SEs **4**–**12** using PASS (prediction of activity spectra for substances; http://www.way2drug.com/passonline/) (accessed on 2 February 2021), which are presented in Table 1. The biological activities against bacteria (0.52 < Pa < 0.58) and fungi (0.66 < Pa < 0.70) clearly suggested that the MDG esters could be significantly potent against phytopathogenic fungi, as compared to that against bacterial pathogens. In addition, the observed antifungal activities are comparable to standard antifungal antibiotic fluconazole (Pa = 0.726).

Having notable antifungal potentiality, we were interested in predicting the antiviral activities of esters **4**–**12**, along with Retrovir and Remdesivir (a drug used for COVID-19), using antiviral predication software (http://crdd.osdd.net/servers/avcpred; accessed on 4 February 2021; Table 2). The introduction of ester groups increased the activity of **4**–**12** against HIV (57–65%) as compared to MDG (41.5%), although less than the HIV drug Retrovir (92.855%). To our surprise, divaleroyl compound **5** was found to have significant potential against HHV (human herpes virus; 91.166%) as compared to Retrovir (28.728%) and Remdesivir (34.473%).

### 2.6. In Vitro Antimicrobial Activities of MDG Esters ***4**–**12***

**Antibacterial activities**. The disc diffusion method [40] was used for the antibacterial activity test against one Gram-positive (*Staphylococcus aureus*) and two Gram-negative (*Escherichia coli* and *Pseudomonas aeruginosa*) bacteria. The antibacterial results are reported as the diameter of the inhibition zone (in mm; Table 3; Figure 4a and Appendix A). These galactopyranoside esters exhibited weak to moderate potentiality against Gram-positive *S. aureus*. However, these SEs did not show any inhibition zone against the Gram-negative organisms.

**Antifungal activities**. It is evident from Table 4 (Figure 4b and Appendix A) that the newly synthesized MDG esters had encouraging antifungal activities. The in vitro antifungal test [41,42] results are very similar to the standard antibiotic fluconazole (Table 3). In many cases, the percentage of the inhibition zone was better than fluconazole. For example, 3,4-di-*O*-octanoate **11** (*78.3%) showed the highest inhibition against *A. fumigatus,* followed by 2,3,4-tri-*O*-octanoate **8** (*76.0%) and 2,3,4-tri-*O*-hexanoate **7** (*71.6%), while the inhibitory activity of the standard antibiotic fluconazole was *65.0%. On the other hand, 2,3,4-tri-*O*-octanoate **8** (*66.7%) exhibited the highest inhibition against *A. niger*, which was very much better than that of the standard antifungal antibiotic fluconazole (37.1%). Hence, the MDG esters showed encouraging antifungal activities, as predicted by PASS predication (Table 1).

### 2.7. Molecular Docking Studies

Molecular docking is the most frequently used technique in current drug development processes to find new hits for a target by minimizing time and expense. As the MDG esters showed promising in vitro antifungal potentiality (Table 1 and Table 4), we were interested in verifying it by docking with a related enzyme, namely sterol 14α-demethylase. In addition, several antifungal drugs (e.g., itraconazole) are reported to possess anti-COVID-19 activities [43]. Considering this observation and in search of novel applications of these MDG compounds, we conducted docking against the SARS-CoV-2 main protease. Here, due to its excellent algorithm to filter out the false-positive ligand by the strict scoring function, XP docking was adopted for our research. Molecular docking of our synthetic compounds was conducted against the fungus enzyme and the main protease of SARS-CoV-2, separately. Before docking simulations, however, the accuracy of docking was validated by re-docking a known reference ligand using the technique of XP docking. In the case of sterol 14α-demethylase (4UYL, *Aspergillus fumigatus*), the reference ligand was VNI (PubChem CID: 49867823) and in the case of the SARS-CoV-2 main protease (6LU7), reference ligand was inhibitor N3 (PubChem CID: 146025593). The reference co-crystal ligand was first separated from the crystal structure and re-docked using the Glide XP protocol. Based on their highest negative XP docking score, we enlisted the compounds after XP docking. The docking scores of sterol 14α-demethylase complexes are shown in Table 5.

In the case of antifungal virtual screening, compounds **8** (−10.551 kcal/mol) and **12** (−10.532 kcal/mol) showed higher XP docking scores than the native ligand VNI (−10.249 kcal/mol). Further, the non-bond interaction of the best complexes along with the reference ligand complex was analyzed to compare the binding mode of our synthetic compounds. Figure 5 represents the non-bond interaction of compound **8**–sterol 14α-demethylase complex, and VNI–sterol 14α-demethylase complex. The reference compound VNI exhibited one conventional hydrogen bond with HIS310, one carbon–hydrogen bond with SER375, two alkyl bonds with LEU125 and LEU503, and ten pi-alkyl bonds with TYR122, PHE229, ALA307, ILE373, LEU503, LEU125, VAL135, VAL150, ALA303, and LEU304 (Figure 5a,b). In contrast, compound **8** exhibited one carbon–hydrogen bond with TYR136, eight alkyl bonds with ALA469, LYS147, VAL150, ILE464, LEU367, LEU473, LEU125, and LEU503, and six pi-alkyl bonds with TYR122, PHE229, PHE234, HIS310, PHE456, and PHE504 (Figure 5c,d). On the other hand, compound **12** exhibited two conventional hydrogen bonds with TYR136 and CYS463, five carbon–hydrogen bonds with ALA307, GLY457, TYR136, HIS461, and ARG462, seven alkyl bonds with ILE373, ILE376, PRO403, LYS147, VAL150, MET300, and LEU503, and five pi-alkyl bonds with TYR122, PHE229, PHE234, HIS310, and PHE504 (Figure 5e,f). On average, compounds **8** and **12** indicated far stronger binding than the native ligand, and even fluconazole and tetracycline. Thus, the docking (binding) scores also supported the in vitro results that MDG esters bear excellent antifungal properties.

The molecular docking results of targeting CYP and ligand molecules indicate that multiple interactions were observed at the active sites of sterol 14α-demethylase; LEU125, LEU503, TYR122, PHE229, ALA307, ILE373, LEU503, LEU125, TYR136, LYS147, ILE464, LEU367, CYS464, ILE376, LYS147, PHE234, and PHE504 residues. The multiple interactions at the active site cavity by the ligand molecules may interfere with the functions of the targeted protein or enzymes [44]. Therefore, multiple trajectories from the molecular dynamics simulation study demonstrate the rigid and stable nature of the docked complexes. Based on the findings, it may be concluded that the ligand molecules may inhibit the CYP but these data need to be validated.

It is interesting to know whether the MDG esters (**4**–**12**) or their hydrolyzed fatty acids (valeric acid, caproic acid, caprylic acid, capric acid, etc.) are responsible for the activity. The lanosterol 14α-demethylase with these acids indicated a much lower value (−3.4 to −4.6 kcal/mol) of binding affinity (Table 5). Again, if hydrolysis occurred in a fungal/bacterial cell wall or membrane, then the resulting MDG sugar must show similar in vitro results, which were not found practically (in fact the use of such MDG as a control showed a variable lower zone of inhibition; Table 3 and Table 4). These observations clearly support that MDG esters are responsible for antimicrobial activities.

The docking scores of synthetic compounds **4**–**12** with the SARS-CoV-2 main protease complexes are presented in Table 6. In the case of the SARS-CoV-2 main protease (6LU7), compounds **4** (−8.976 kcal/mol) and **5** (−8.464 kcal/mol) showed higher XP docking scores than the native inhibitor N3 (−8.151 kcal/mol).

To further understand the binding behavior of the protein–ligand complex, we analyzed the non-bond interaction and compared the non-bond interaction of our synthetic compounds with the reference ligand inhibitor N3, according to bond categories [45]. Figure 6 represents the non-bond interaction of the compound **4**–6LU7 complex and the compound **5**–6LU7 main protease complex, along with the reference inhibitor N3–6LU7 complex. Here, the reference inhibitor N3 formed four conventional hydrogen bonds with GLU166, GLN189, PHE140, and THR190, five carbon–hydrogen bonds with HIS163, MET165, GLU166, ARG188, and HIS164, two amide-pi-stacked bonds with LEU141 and MET165, two alkyl bonds with CYS145 and PRO168, two pi-alkyl bonds with HIS163 and MET49, and a pi-sulfur bond with MET165 (Figure 6a,b). In contrast, compound **4** formed two conventional hydrogen bonds with GLN189 and THR190, five carbon–hydrogen bonds with PRO168, GLN189, THR190, ARG188, and GLU166, an alkyl bond with MET49, and a pi-alkyl bond with HIS41 (Figure 6c,d). On the other hand, compound **5** possessed two conventional hydrogen bonds with GLN189 and THR190, five carbon–hydrogen bonds with PRO168, GLN189, THR190, GLU166, and ARG188, and two alkyl bonds with MET49 and PRO52 (Figure 6e,f). From the non-bond interaction analysis, it was revealed that the binding site of the synthetic compounds and the reference ligand N3 was the same.

### 2.8. Molecular Dynamics Simulation

The molecular dynamics simulation study was conducted to understand the structural stability of the complexes. The root mean square deviations (RMSDs) of the C-alpha atoms were evaluated for both complexes. The 4UYL complexes’ control and the other complexes had similar RMSD patterns, wherein complex **12** had a lower RMSD than the other complexes (Figure 7a). The main protease of the SARS-CoV-2 complexes indicates that all the complexes exhibit an initial upper trend, which might be responsible for the flexible nature. Therefore, the control complexes demonstrated a higher RMSD compared to the other complexes, which indicates the complexes’ flexible nature. Moreover, compound **4** and compound **5** complexes had a lower RMSD compared to the control in the whole simulation times (Figure 7f). This trend defines the stable nature of the docked complexes. The average RMSD of all the complexes was below 2.5 Å, which indicates the overall stability of the complexes.

The solvent-accessible surface area of the complexes indicates the changes in the surface volumes of the complexes, where a higher SASA is related to the extension of the surface, whereas the lower SASA is related to the truncated nature of the complexes. Figure 7b indicates that for the 4UYL complexes, the control exhibits a lower SASA, which is related to the more truncated nature than the other complexes. The main protease complexes exhibit a similar SASA (Figure 7g) for all of the complexes compared to the control.

The radius of gyrations (Rg) of the complexes indicates the labile nature of the complexes, where the lower Rg is related to the stable nature of the complexes and the higher Rg relates to the more mobile nature. Figure 7c indicates that the 4UYL complexes had a similar Rg pattern and did not over fluctuate across the simulation periods. Figure 7h indicates that all of the complexes of the main protease also had similar Rg patterns and did not fluctuate across the simulation times.

Finally, the hydrogen bond of a simulation system defines the stable nature of the complexes. Figure 7d,i indicate that the complexes from 4UYL and the main protease had a similar stable nature. Therefore, the binding free energy was calculated by MM-PBSA methods, where positive energy indicates the positive binding of the complexes [46]. The 4UYL compound **8** complexes had higher binding free energy than the other complexes, which indicates more favorable binding (Figure 7e). Figure 7j indicates that compound **4** from the main protease complexes had higher binding energy than the other complexes, although they exhibited almost similar MM-PBSA scores in the simulations.

### 2.9. In Silico Inhibition Constant (Ki) Analysis

When selecting a potent inhibitor for a particular receptor, we cannot rule out the value of IC_50_. Therefore, the present research also included the theoretical IC_50_ of the best ligands and the reference ligands for sterol 14α-demethylase (4UYL) and SARS-CoV-2 M^pro^ (6LU7) by in silico methods [47]. The results (Table 7) showed a good inhibition constant (µM) in both cases.

### 2.10. ADMET Studies

Pharmacokinetic properties such as absorption, distribution, metabolism, excretion, and toxicity (ADMET) calculations of galactopyranosides **3**–**12** from pkCSM are shown in Table 8.

**Absorption**. As observed in Table 8, compounds **3** and **4** are water-soluble, while compounds **5**, **6**, **8**, **9,** and **12** are slightly water-soluble, and **7**, **10,** and **11** are water-insoluble (lipid-soluble). They all are also Caco2-permeable, except compounds **3**, **4,** and **5,** which are poorly permeable. The permeability of the Caco-2 cell line, which is made up of human epithelial colorectal adenocarcinoma cells, is used to predict the absorption of orally delivered drugs. Except for the highly hydrophilic non-ester **3**, all the compounds have a high % of absorption in the intestine. Skin permeability is moderate in all the derivatives (except **5**, which has relatively low skin permeability). Transdermal drug design considers skin-permeable compounds. The ATP-binding cassette (ABC), also known as P-glycoprotein, functions as a biological barrier by extruding xenobiotics from cells. It plays a crucial function in drug transportation in various tissues by restricting drug cellular absorption from the blood circulation into the brain and from the intestinal lumen into epithelial cells, rather than enhancing drug excretion from hepatocytes and renal tubules into the surrounding luminal space. Drugs that stimulate P-glycoprotein can lower the bioavailability of other drugs. P-glycoprotein inhibitors increase the bioavailability of drugs that are vulnerable to P-glycoprotein inhibition [48]. P-glycoprotein does not interact with any of the drugs in this study and compounds **3**–**6** do not inhibit P-glycoprotein (I and II). P-glycoprotein was inhibited by chemicals **7**–**12**.

**Distribution**. The volume of distribution (VD) concept is used to anticipate whether a drug’s distribution in blood plasma and tissue is uniform or not. The higher the VD value, the more the drug is dispersed in tissue as opposed to plasma [49]. Except for **6** and **9**, all the compounds have a low VD, though all of the derivatives have equal or better VD values than fluconazole. The unbound fraction in plasma is a crucial predictor of drug effectiveness, since only the unbound drug can interact with pharmacological target proteins such as receptors, channels, and enzymes, and is capable of moving between plasma and tissues. Surprisingly, all the compounds have a high fraction unbound (Fu) value, except for compounds **8**, **11**, and **12**, which have a lower Fu value despite having a higher Fu value than fluconazole. A similar study shows that all compounds are unable to pass through the CNS and are extremely poorly permeable to the Blood–Brain Barrier (BBB), with the exception of compound **3**, which is significant for reducing unexpected adverse effects.

**Metabolism**. Cytochrome P450 is a metabolism enzyme in the body that oxidizes xenobiotics to make them easier to excrete. Many drugs deactivate cytochrome P450, and certain drugs can activate it [13]. The multiple isoforms of cytochrome P450 substrate/inhibitor are used in in silico algorithms to predict drug metabolism. Except for compounds **6**–**12**, which can only interact with CYP3A4, none of the drugs interact with cytochrome P450 isoforms CYP1A2, CYP2C19, CYP2C9, CYP2D6, or CYP3A4. These findings indicate that these novel compounds are not maximal cytochrome P450 isoform metabolites. As a result, all the compounds have a good metabolism and are likely to be therapeutic molecules.

**Excretion**. Organic cation transporter 2 (OCT2) is a transport protein located in the kidney. It is a renal uptake carrier that is vital for medication disposal and renal clearance. OCT2 inducer candidates boost OCT2 activity (no OCT2 inducers have been found), while OCT2 inhibitors decrease OCT2 action [50]. Except for compounds **7**, **10**, and **11**, the majority of these synthetic chemicals do not interact with OCT2. The total clearance of the drugs is also calculated, which can help with drug dosing.

**Toxicity**. Toxicity analysis revealed that all of the synthesized chemicals were safe, with the exception of compound **6**, which was shown to be hepatotoxic (Table 8). The AMES toxicity, hepatotoxicity, and skin sensitization in silico assessments were performed. hERG (human ether-a-go-go-gene) potassium channels are required for proper electrical activity in the heart. Long QT syndrome is caused by hereditary abnormalities in the hERG gene or inhibition of hERG channels by a set of drugs [51]. Surprisingly, complete synthesized compounds are found to be free of interactions with hERG I and hERG II. Understanding a possible compound’s hazardous potency is critical. To measure the relative toxicity of different compounds, a common acute toxicity calculation is utilized; this is the lethal dose value (LD_50_). The LD_50_ is the amount of a chemical that, when administered all at once, kills 50% of a set of test animals. In Table 8, we have listed the LD_50_ values in mol/kg units. Exposure to low–moderate dosages of chemicals over lengthy periods of time is a major risk in numerous treatment approaches. Chronic research aims to classify the lowest dose of a substance that causes an unfavorable impact (LOAEL) and the greatest dose that has no adverse effects (NOAEL). The LOAEL value has been expressed as mg/kg body weight/day. The maximum doses that could be tolerated were also calculated. Several chemicals had to be withdrawn from pre-clinical trials and even the pharmaceutical market because of toxicity exposure. All synthesized molecules **4**–**12** are safe and nontoxic in this regard.

### 2.11. Drug-Likeness Results

All physicochemical properties of the derivatives, such as the number of heavy atoms, number of aromatic heavy atoms, fraction Csp3, rotatable bonds, number of acceptors and donors of hydrogen bonds, molar refractivity (MR), and topological polar surface area (TPSA) were considered for drug-likeness (Table 9).

Various reports have shown that the number of heavy atoms should be 20 to 70, the Csp3 fraction should be >0.25, the rotatable bonds should be ≤10, the number of acceptors of H-bonds should be ≤10, the number of donors of H-bonds should be ≤5, the MR should be 40 to 130, and the TPSA should be ≤140 Å^2^ [52]. Based on these physicochemical properties, many scientists and pharmaceutical companies have designed different computational filters or protocols, such as Lipinski’s rule of five, the Ghose filter, the Veber filter, and the filters of Egan and Muegge. Further, we considered the computational filters of Lipinski, Ghose, Veber, Egan, and Muegge to evaluate our synthetic compounds. In the case of Lipinski’s rule of five, five compounds showed no violation, while compounds **7**, **8,** and **10**–**12** showed only one violation. Hence, we considered another filter, where the maximum compounds satisfied the drug-likeness evaluation, and the bioavailability score of all the compounds satisfied the drug-likeness tests.

## 3. Materials and Methods

### 3.1. General Methods and Instruments

All chemicals were purchased as reagent grade and used without further purification. Evaporations were conducted below 40 °C in a Buchi rotary evaporator (R-100, Switzerland) under reduced pressure. Thin-layer chromatography (TLC) was performed on the Kieselgel GF_254_ plate and the spots were observed by spraying the plates with 1% H_2_SO_4_ in methanol and heating the plate at 150–200 °C until coloration took place. Column chromatography (CC) was performed with silica gel G_60_. The solvent system employed for the TLC and CC was chloroform/methanol and/or *n*-hexane/ethyl acetate in different proportions. Melting points were determined in an electrothermal melting point apparatus and are uncorrected. FT-IR spectra were recorded on an FT-IR spectrophotometer (Shimadzu, IR Prestige-21, Kyoto, Japan) in the KBr technique. ^1^H (400 MHz) and ^13^C (100 MHz) NMR spectra were recorded in CDCl_3_ solution using a tunable multinuclear probe (Bruker DPX-400 spectrometer, Billerica, USA). Chemical shifts were reported in δ unit (ppm), with reference to TMS as an internal standard and *J* values are shown in Hz. Elemental analyses were conducted with a C/H analyzer.

### 3.2. Synthesis

**Methyl 6-*O*-valeroyl-α-****D-galactopyranoside (4):** To a cooled (0 °C) stirred mixture of methyl α-D-galactopyranoside (MDG, **3**) (2.0 g, 10.299 mmol) and pyridine (4 mL), valeroyl chloride (pentanoyl chloride) (1.262 g, 10.297 mmol) was added slowly. The reaction mixture was stirred at this temperature for 12 h and then for 4 h at room temperature when TLC indicated the complete conversion of the starting compound into a faster-moving product(s) (*R*_f_ = 0.40, chloroform/methanol, 5:1). The usual workup followed by CC with chloroform/methanol (10:1, *v*/*v*) provided pure compound **4** (1.405 g, 49%) as brownish solid, mp 95–96 °C. *R*_f_ = 0.40 (chloroform/methanol = 5/1). FT-IR (KBr): 3200–3450 (br, OH), 1739 (CO), and 1048 cm^−1^ (pyranose ring). ^1^H NMR (400 MHz, CDCl_3_): δ_H_ 4.92 (d, *J* = 3.2 Hz, 1H, H-1), 4.36 (dd, *J* = 11.6 and 4.4 Hz, 1H, H-6), 4.27 (dd, *J* = 11.6 and 6.4 Hz, 1H, H-6′), 3.78–3.98 (m, 4H, H-2, H-3, H-4, and H-5), 3.42 (s, 3H, OC*H*_3_), 3.21–3.38 (br s, 3H, 3×O*H*), 2.35 [t, *J* = 7.6 Hz, 2H, CH_3_(CH_2_)_2_C*H*_2_CO], 1.58–1.69 (m, 2H, CH_3_CH_2_C*H*_2_CH_2_CO), 1.27–1.43 [m, 2H, CH_3_C*H*_2_(CH_2_)_2_CO], and 0.92 [t, 3H, *J* = 7.2 Hz, C*H*_3_(CH_2_)_3_CO]. ^13^C NMR (100 MHz, CDCl_3_): δ_C_ 173.9 [CH_3_(CH_2_)_3_*C*O], 99.5 (C-1), 71.3, 70.7, 68.3 (C-2/C-3/C-4), 68.1 (C-6), 63.2 (C-5), 55.4 (O*C*H_3_), 71.5, 70.5, 70.4 (C-2/C-3/C-4), 67.7 (C-5), 63.8 (C-6), 54.8 (O*C*H_3_), 33.9 [CH_3_(CH_2_)_2_*C*H_2_CO], 27.0 (CH_3_CH_2_*C*H_2_CH_2_CO), 22.2 [CH_3_*C*H_2_(CH_2_)_2_CO], and 13.6 [*C*H_3_(CH_2_)_3_CO]. Anal. Calcd. for C_12_H_22_O_7_ (278.30): C, 51.79; H, 7.97. Found: C, 51.86; H, 7.95.

**Methyl 2,6-di-*O*-valeroyl-α-****D-galactopyranoside (5) and methyl 6-*O*-valeroyl-α-****D-galactopyranoside (4): DMAP catalyzed technique**: To a solution of MDG (3) (2.0 g, 10.299 mmol) in dry pyridine (4 mL), valeroyl chloride (1.262 g, 10.297 mmol) was added slowly at 0 °C, followed by the addition of DMAP (~30 mg). Stirring was continued at this temperature for 2 h and then overnight at 25 °C. TLC indicated the formation of two faster-moving products, having *R*_f_ = 0.59 and 0.40 (chloroform/methanol = 5/1, *v*/*v*). The usual workup and an initial CC elution with chloroform/methanol (15/1) furnished the higher *R*_f_ containing 2,6-di-*O*-pentanoate 5 (2.09 g, 56%) as a colorless solid, mp 58–59 °C. *R*_f_ = 0.59 (chloroform/methanol = 5/1). FT-IR (KBr): 3250–3570 (br, OH), 1736, 1735 (CO), and 1044 cm^−1^ (pyranose ring). ^1^H NMR (400 MHz, CDCl_3_): δ_H_ 5.03 (dd, *J* = 10.2 and 3.6 Hz, 1H, H-2), 4.92 (d, *J* = 3.6 Hz, 1H, H-1), 4.45 (dd, *J* = 11.6 and 6.0 Hz, 1H, H-6), 4.24 (dd, *J* = 11.6 and 7.0 Hz, 1H, H-6′), 3.96–4.04 (m, 3H, H-3, H-4, and H-5), 3.40 (s, 3H, OC*H*_3_), 2.42 [t, *J* = 7.6 Hz, 2H, CH_3_(CH_2_)_2_C*H*_2_CO], 2.36 [t, *J* = 7.6 Hz, 2H, CH_3_(CH_2_)_2_C*H*_2_CO], 1.59–1.69 (m, 4H, 2×CH_3_CH_2_C*H*_2_CH_2_CO), 1.34–1.40 [m, 4H, 2×CH_3_C*H*_2_(CH_2_)_2_CO], 0.95 [t, *J* = 7.2 Hz, 3H, C*H*_3_(CH_2_)_3_CO], and 0.92 [t, *J* = 7.2 Hz, 3H,C*H*_3_(CH_2_)_3_CO]. ^13^C NMR (100 MHz, CDCl_3_): δ_C_ 174.4, 173.9 [2×CH_3_(CH_2_)_3_*C*O], 97.5 (C-1), 71.5, 69.3, 68.1 (C-2/C-3/C-4), 67.5 (C-5), 62.8 (C-6), 55.3 (O*C*H_3_), 34.0, 33.9 [2×CH_3_(CH_2_)_2_*C*H_2_CO], 27.0, 26.9 (2×CH_3_CH_2_*C*H_2_CH_2_CO), 22.2, 22.1 [2×CH_3_*C*H_2_(CH_2_)_2_CO], and 13.6(2) [2×*C*H_3_(CH_2_)_3_CO]. The assignments of the signals of this compound were established by analyzing its COSY and HMBC experiments. Anal. Calcd. for C_17_H_30_O_8_ (362.42): C, 56.34; H, 8.34. Found: C, 56.40; H, 8.37.

Further CC elution with chloroform/methanol = 12/1 (*v*/*v*) provided slower-moving (*R*_f_ = 0.40) component **4** (0.831 g, 29%) as a brownish solid. Its FT-IR, ^1^H, and ^13^C NMR spectra were indistinguishable from those prepared in the earlier step by the direct valeroylation.

*General procedure for 2,3,4-tri-O-acylation of compound **4** and 2,4-di-O-acylation of compound **5** using the direct method.* The necessary acyl halide (3.3 or 2.2 eq.) was slowly added to a solution of **4** or **5** (0.1 g) in dry pyridine (1 mL) at a low temperature (0 °C). In addition, a catalytic amount of DMAP was added to the reaction mixture. After 30 min, it was stirred for 11–16 h at 25 °C. The reaction mixture was quenched with ice water, followed by extraction with DCM (5 × 3 mL). The organic layer was washed with 5% HCl, followed by aqueous NaHCO_3_ and brine wash. The organic layer free from pyridine was dried (MgSO_4_) and evaporated to leave a viscous syrup, which was purified by CC (elution with *n*-hexane/ethyl acetate) and the desired 2,3,4-tri- and 2,4-di-*O*-acyl products, respectively, were obtained.

**Methyl 2,3,4-tri-*O*-acetyl-6-*O*-valeroyl-α-****D-galactopyranoside (6):** Thick syrup; yield 92%; *R*_f_ = 0.53 (*n*-hexane/EA = 4/1). FT-IR (KBr): 1755, 1754, 1745, 1743 (CO), and 1051 cm^−1^ (pyranose ring). ^1^H NMR (400 MHz, CDCl_3_): δ_H_ 5.45 (d, *J* = 3.2 Hz, 1H, H-4), 5.36 (dd, *J* = 11.0 and 3.2 Hz, 1H, H-3), 5.18 (dd, *J* = 11.0 and 3.6 Hz, 1H, H-2), 5.00 (d, *J* = 3.6 Hz, 1H, H-1), 4.20 (apparent t, *J* = 6.4 Hz, 1H, H-5), 4.10–4.14 (m, 2H, H-6 and H-6′), 3.42 (s, 3H, OC*H*_3_), 2.33 [t, *J* = 7.2 Hz, 2H, CH_3_(CH_2_)_2_C*H*_2_CO], 2.15 (s, 3H, C*H*_3_CO), 2.10 (s, 3H, C*H*_3_CO), 1.99 (s, 3H, C*H*_3_CO), 1.53–1.64 (m, 2H, CH_3_CH_2_C*H*_2_CH_2_CO), 1.31–1.38 [m, 2H, CH_3_C*H*_2_(CH_2_)_2_CO], and 0.91 [t, *J* = 7.6 Hz, 3H, C*H*_3_(CH_2_)_3_CO]. ^13^C NMR (100 MHz, CDCl_3_): δ_C_ 173.2 [CH_3_(CH_2_)_3_*C*O], 170.4, 170.1, 169.9 (3×CH_3_*C*O), 97.2 (C-1), 68.2, 67.6, 66.2 (C-2/C-3/C-4), 66.2 (C-5), 61.6 (C-6), 55.5 (O*C*H_3_), 33.7 [CH_3_(CH_2_)_2_*C*H_2_CO], 26.8 (CH_3_CH_2_*C*H_2_CH_2_CO), 22.2 [2×CH_3_*C*H_2_(CH_2_)_2_CO], 20.8, 20.6(2) (3×*C*H_3_CO), and 13.6 [*C*H_3_(CH_2_)_3_CO]. The assignments of the signals of this compound were established by analyzing its COSY, HSQC, and HMBC experiments. Anal. Calcd. for C_18_H_28_O_10_ (404.41): C, 53.46; H, 6.98. Found: C, 53.50; H, 6.96.

**Methyl 2,3,4-tri-*O*-hexanoyl-6-*O*-valeroyl-α-****D-galactopyranoside (7):** Syrup; yield 81%; *R*_f_ = 0.61 (*n*-hexane/EA = 4/1). FT-IR (KBr): 1742, 1738, 1718, 1716 (CO), and 1065 cm^−1^ (pyranose ring). ^1^H NMR (400 MHz, CDCl_3_): δ_H_ 5.49 (d, *J* = 2.8 Hz, 1H, H-4), 5.41 (dd, *J* = 10.8 and 3.2 Hz, 1H, H-3), 5.17 (dd, *J* = 10.8 and 3.6 Hz, 1H, H-2), 5.01 (d, *J* = 3.6 Hz, 1H, H-1), 4.19–4.24 (m, 1H, H-5), 4.09–4.15 (m, 2H, H-6 and H-6′), 3.42 (s, 3H, OC*H*_3_), 2.28–2.43, 2.18–2.22 [2×m, 8H, CH_3_(CH_2_)_2_C*H*_2_CO and 3×CH_3_(CH_2_)_3_C*H*_2_CO], 1.54–1.71 (m, 8H, CH_3_CH_2_C*H*_2_CH_2_CO and 3×CH_3_(CH_2_)_2_C*H*_2_CH_2_CO), 1.24–1.40 [m, 14H, br CH_3_C*H*_2_(CH_2_)_2_CO and 3×CH_3_(C*H*_2_)_2_CH_2_CH_2_CO], and 0.87–0.95 [m, 12H, C*H*_3_(CH_2_)_3_CO and 3×C*H*_3_(CH_2_)_4_CO]. ^13^C NMR (100 MHz, CDCl_3_): δ_C_ 173.2(2), 172.9, 172.5 [CH_3_(CH_2_)_3_*C*O and 3×CH_3_(CH_2_)_4_*C*O], 97.4 (C-1), 68.1, 68.0, 67.4 (C-2/C-3/C-4), 66.4 (C-5), 61.7 (C-6), 55.5 (O*C*H_3_), 34.1, 34.0, 33.7, 33.6 [CH_3_(CH_2_)_2_*C*H_2_CO and 3×CH_3_(CH_2_)_3_*C*H_2_CO], 31.2(2), 31.1 [3×CH_3_(CH_2_)_2_*C*H_2_CH_2_CO], 26.8 (CH_3_CH_2_*C*H_2_CH_2_CO), 24.7, 24.3, 24.2 [3×CH_3_CH_2_*C*H_2_*(*CH_2_)_2_CO], 22.3(2), 22.2, 22.1 [CH_3_*C*H_2_(CH_2_)_2_CO and 3×CH_3_*C*H_2_(CH_2_)_3_CO], 13.8(3), and 13.6 [*C*H_3_(CH_2_)_3_CO and 3×*C*H_3_(CH_2_)_4_CO]. Anal. Calcd. for C_30_H_52_O_10_ (572.73): C, 62.91; H, 9.15. Found: C, 62.89; H, 9.20.

**Methyl 2,3,4-tri-*O*-octanoyl-6-*O*-valeroyl-α-****D-galactopyranoside (8):** Semi-solid; yield 82%; *R*_f_ = 0.66 (*n*-hexane/EA = 4/1). FT-IR (KBr): 1742, 1741, 1709, 1705 (CO), and 1055 cm^−1^ (pyranose ring). ^1^H NMR (400 MHz, CDCl_3_): δ_H_ 5.48 (d, *J* = 2.8 Hz, 1H, H-4), 5.38 (dd, *J* = 10.8 and 3.2 Hz, 1H, H-3), 5.16 (dd, *J* = 11.2 and 3.2 Hz, 1H, H-2), 5.00 (d, *J* = 3.6 Hz, 1H, H-1), 4.18–4.23 (m, 1H, H-5), 4.05–4.14 (m, 2H, H-6 and H-6′), 3.41 (s, 3H, OC*H*_3_), 2.28–2.43, 2.17–2.21 [2×m, 8H, CH_3_(CH_2_)_2_C*H*_2_CO and 3×CH_3_(CH_2_)_5_C*H*_2_CO], 1.56–1.70 (m, 8H, CH_3_CH_2_C*H*_2_CH_2_CO and 3×CH_3_(CH_2_)_4_C*H*_2_CH_2_CO), 1.23–1.39 [br m, 26H, CH_3_C*H*_2_(CH_2_)_2_CO and 3×CH_3_(C*H*_2_)_4_CH_2_CH_2_CO], and 0.86–0.96 [m, 12H, C*H*_3_(CH_2_)_3_CO and 3×C*H*_3_(CH_2_)_6_CO]. ^13^C NMR (100 MHz, CDCl_3_): δ_C_ 173.2(2), 173.9, 172.6 [CH_3_(CH_2_)_3_*C*O and 3×CH_3_(CH_2_)_6_*C*O], 97.3 (C-1), 68.1, 68.0, 67.4 (C-2/C-3/C-4), 66.4 (C-5), 61.7 (C-6), 55.4 (O*C*H_3_), 34.1(3), 34.0 [CH_3_(CH_2_)_2_*C*H_2_CO and 3×CH_3_(CH_2_)_5_*C*H_2_CO], 31.6(3) [3×CH_3_(CH_2_)_4_*C*H_2_CH_2_CO], 29.1(3) [3×CH_3_(CH_2_)_3_*C*H_2_(CH_2_)_2_CO], 28.8(3) [3×CH_3_(CH_2_)_2_*C*H_2_(CH_2_)_3_CO], 26.8 (CH_3_CH_2_*C*H_2_CH_2_CO), 24.6(3) [3×CH_3_CH_2_*C*H_2_*(*CH_2_)_4_CO], 22.5(3), 22.2 [CH_3_*C*H_2_(CH_2_)_2_CO and 3×CH_3_*C*H_2_(CH_2_)_5_CO], 14.0(3), and 13.6 [*C*H_3_(CH_2_)_3_CO and 3×*C*H_3_(CH_2_)_6_CO]. Anal. Calcd. for C_36_H_64_O_10_ (656.89): C, 65.82; H, 9.82. Found: C, 65.89; H, 9.85.

**Methyl 3,4-di-*O*-acetyl-2,6-di-*O*-valeroyl-α-****D-galactopyranoside (9):** Thick syrup; yield 92%; *R*_f_ = 0.60 (*n*-hexane/EA = 4/1). FT-IR (KBr): 1737, 1733, 1729, 1721 (CO), and 1070 cm^−1^ (pyranose ring). ^1^H NMR (400 MHz, CDCl_3_): δ_H_ 5.45 (d, *J* = 2.8 Hz, 1H, H-4), 5.37 (dd, *J* = 10.8 and 3.2 Hz, 1H, H-3), 5.16 (dd, *J* = 10.8 and 3.6 Hz, 1H, H-2), 5.00 (d, *J* = 3.6 Hz, 1H, H-1), 4.17–4.23 (m, 1H, H-5), 4.10–4.14 (m, 2H, H-6 and H-6′), 3.41 (s, 3H, OC*H*_3_), 2.36 [t, *J* = 7.6 Hz, 2H, CH_3_(CH_2_)_2_C*H*_2_CO], 2.31 [t, *J* = 7.2 Hz, 2H, CH_3_(CH_2_)_2_C*H*_2_CO], 2.15 (s, 3H, C*H*_3_CO), 1.98 (s, 3H, C*H*_3_CO), 1.56–1.64 (m, 4H, 2×CH_3_CH_2_C*H*_2_CH_2_CO), 1.27–1.39 [m, 4H, 2×CH_3_C*H*_2_(CH_2_)_2_CO], and 0.91 [t, *J* = 7.2 Hz, 6H, 2×C*H*_3_(CH_2_)_3_CO]. ^13^C NMR (100 MHz, CDCl_3_): δ_C_ 173.2(2) [2×CH_3_(CH_2_)_3_*C*O], 170.2, 169.9 (2×CH_3_*C*O), 97.3 (C-1), 68.2, 67.9, 67.6 (C-2/C-3/C-4), 66.2 (C-5), 61.0 (C-6), 55.4 (O*C*H_3_), 33.9, 33.7 [2×CH_3_(CH_2_)_2_*C*H_2_CO], 27.0, 26.8 (2×CH_3_CH_2_*C*H_2_CH_2_CO), 22.2, 22.1 [2×CH_3_*C*H_2_(CH_2_)_2_CO], 20.6, 20.5 (2×*C*H_3_CO), and 13.6(2) [2×*C*H_3_(CH_2_)_3_CO]. The assignments of the signals of this compound were established by analyzing its COSY, HSQC, and HMBC experiments. Anal. Calcd. for C_21_H_34_O_10_ (446.49): C, 56.49; H, 7.68. Found: C, 56.55; H, 7.66.

**Methyl 3,4-di-*O*-hexanoyl-2,6-di-*O*-valeroyl-α-****D-galactopyranoside (10):** Clear oil; yield 87%; *R*_f_ = 0.52 (*n*-hexane/EA = 4/1). FT-IR (KBr): 1749, 1747, 1719, 1711 (CO), and 1050 cm^−1^ (pyranose ring). ^1^H NMR (400 MHz, CDCl_3_): δ_H_ 5.48 (d, *J* = 2.8 Hz, 1H, H-4), 5.38 (dd, *J* = 10.8 and 3.2 Hz, 1H, H-3), 5.17 (dd, *J* = 10.8 and 3.6 Hz, 1H, H-2), 5.00 (d, *J* = 3.6 Hz, 1H, H-1), 4.17–4.23 (m, 1H, H-5), 4.05–4.15 (m, 2H, H-6 and H-6′), 3.41 (s, 3H, OC*H*_3_), 2.26–2.43, 2.18–2.23 [2×m, 8H, 2×CH_3_(CH_2_)_2_C*H*_2_CO and 2×CH_3_(CH_2_)_3_C*H*_2_CO], 1.50–1.69 (m, 8H, 2×CH_3_CH_2_C*H*_2_CH_2_CO and 2×CH_3_(CH_2_)_2_C*H*_2_CH_2_CO), 1.26–1.41 [m, 12H, 2×CH_3_C*H*_2_(CH_2_)_2_CO and 2×CH_3_(C*H*_2_)_2_CH_2_CH_2_CO], 0.94 [t, *J* = 7.2 Hz, 6H, 2×C*H*_3_], and 0.89 [t, *J* = 6.8 Hz, 6H, 2×C*H*_3_]. ^13^C NMR (100 MHz, CDCl_3_): δ_C_ 173.3, 173.2, 172.6(2) [2×CH_3_(CH_2_)_3_*C*O and 2×CH_3_(CH_2_)_4_*C*O], 97.3 (C-1), 68.0, 67.9, 67.4 (C-2/C-3/C-4), 66.4 (C-5), 61.7 (C-6), 55.4 (O*C*H_3_), 34.0, 33.9, 33.8, 33.7 [2×CH_3_(CH_2_)_2_*C*H_2_CO and 2×CH_3_(CH_2_)_3_*C*H_2_CO], 31.2(2) [2×CH_3_(CH_2_)_2_*C*H_2_CH_2_CO], 27.0, 26.8 (2×CH_3_CH_2_*C*H_2_CH_2_CO), 24.6, 24.4 [2×CH_3_CH_2_*C*H_2_*(*CH_2_)_2_CO], 22.3(3), 22.2 [CH_3_*C*H_2_(CH_2_)_2_CO and 3×CH_3_*C*H_2_(CH_2_)_3_CO], 13.8(3), and 13.6 [2×*C*H_3_(CH_2_)_3_CO and 2×*C*H_3_(CH_2_)_4_CO]. Anal. Calcd. for C_29_H_50_O_10_ (558.70): C, 62.34; H, 9.02. Found: C, 62.38; H, 9.08.

**Methyl 3,4-di-*O*-octanoyl-2,6-di-*O*-valeroyl-α-****D-galactopyranoside (11):** Syrup; yield 84%; *R*_f_ = 0.56 (*n*-hexane/EA = 4/1). FT-IR (KBr): 1748, 1747, 1714, 1713 (CO), and 1051 cm^−1^ (pyranose ring). ^1^H NMR (400 MHz, CDCl_3_): δ_H_ 5.47 (d, *J* = 2.8 Hz, 1H, H-4), 5.37 (dd, *J* = 10.8 and 3.2 Hz, 1H, H-3), 5.15 (dd, *J* = 10.8 and 3.2 Hz, 1H, H-2), 4.99 (d, *J* = 3.6 Hz, 1H, H-1), 4.18–4.22 (m, 1H, H-5), 4.05–4.15 (m, 2H, H-6 and H-6′), 3.40 (s, 3H, OC*H*_3_), 2.28–2.42, 2.16–2.21 [2×m, 8H, 2×CH_3_(CH_2_)_2_C*H*_2_CO and 2×CH_3_(CH_2_)_5_C*H*_2_CO], 1.51–1.67 (m, 8H, 2×CH_3_CH_2_C*H*_2_CH_2_CO and 2×CH_3_(CH_2_)_4_C*H*_2_CH_2_CO), 1.22–1.39 [br m, 20H, 2×CH_3_C*H*_2_(CH_2_)_2_CO and 2×CH_3_(C*H*_2_)_4_CH_2_CH_2_CO], and 0.86–0.94 [m, 12H, 2×C*H*_3_(CH_2_)_3_CO and 2×C*H*_3_(CH_2_)_6_CO]. ^13^C NMR (100 MHz, CDCl_3_): δ_C_ 173.3, 173.2, 172.9, 172.6 [2×CH_3_(CH_2_)_3_*C*O and 2×CH_3_(CH_2_)_6_*C*O], 97.3 (C-1), 68.1, 68.0, 67.4 (C-2/C-3/C-4), 66.3 (C-5), 61.7 (C-6), 55.4 (O*C*H_3_), 34.1 (3), 34.0 [2×CH_3_(CH_2_)_2_*C*H_2_CO and 2×CH_3_(CH_2_)_5_*C*H_2_CO], 31.2(2) [2×CH_3_(CH_2_)_4_*C*H_2_CH_2_CO], 29.0 [2×CH_3_(CH_2_)_3_*C*H_2_(CH_2_)_2_CO], 28.7(2) [2×CH_3_(CH_2_)_2_*C*H_2_(CH_2_)_3_CO], 27.0, 26.8 (2×CH_3_CH_2_*C*H_2_CH_2_CO), 24.6(3) [2×CH_3_CH_2_*C*H_2_*(*CH_2_)_4_CO], 22.5(3), 22.2 [2×CH_3_*C*H_2_(CH_2_)_2_CO and 2×CH_3_*C*H_2_(CH_2_)_5_CO], 14.0(2), 13.6, and 13.5 [2×*C*H_3_(CH_2_)_3_CO and 2×*C*H_3_(CH_2_)_6_CO]. Anal. Calcd. for C_33_H_58_O_10_ (614.81): C, 64.47; H, 9.51. Found: C, 64.51; H, 9.55.

**Methyl 3,4-di-*O*-decanoyl-2,6-di-*O*-valeroyl-α-****D-galactopyranoside (12):** Syrup; yield 82%; *R*_f_ = 0.67 (*n*-hexane/EA = 4/1). FT-IR (KBr): 1751, 1747, 1715, 1710 (CO), and 1052 cm^−1^ (pyranose ring). ^1^H NMR (400 MHz, CDCl_3_): δ_H_ 5.48 (d, *J* = 2.8 Hz, 1H, H-4), 5.38 (dd, *J* = 10.8 and 3.2 Hz, 1H, H-3), 5.17 (dd, *J* = 10.8 and 3.6 Hz, 1H, H-2), 5.00 (d, *J* = 3.6 Hz, 1H, H-1), 4.18–4.23 (m, 1H, H-5), 4.06–4.16 (m, 2H, H-6 and H-6′), 3.42 (s, 3H, OC*H*_3_), 2.29–2.43, 2.18–2.23 [2×m, 8H, 2×CH_3_(CH_2_)_2_C*H*_2_CO and 2×CH_3_(CH_2_)_7_C*H*_2_CO], 1.50–1.69 (m, 8H, 2×CH_3_CH_2_C*H*_2_CH_2_CO and 2×CH_3_(CH_2_)_6_C*H*_2_CH_2_CO), 1.20–1.42 [br m, 28H, 2×CH_3_C*H*_2_(CH_2_)_2_CO and 2×CH_3_(C*H*_2_)_6_CH_2_CH_2_CO], and 0.86–0.97 [m, 12H, 2×C*H*_3_(CH_2_)_3_CO and 2×C*H*_3_(CH_2_)_8_CO]. ^13^C NMR (100 MHz, CDCl_3_): δ_C_ 173.2, 173.1, 172.9, 172.5 [2×CH_3_(CH_2_)_3_*C*O and 2×CH_3_(CH_2_)_8_*C*O], 97.3 (C-1), 68.0 (2), 67.4 (C-2/C-3/C-4), 66.4 (C-5), 61.7 (C-6), 55.4 (O*C*H_3_), 34.1, 34.0(2) [2×CH_3_(CH_2_)_7_*C*H_2_CO], 33.8(3), 33.7 [2×CH_3_(CH_2_)_2_*C*H_2_CO and 2×CH_3_(CH_2_)_6_*C*H_2_CH_2_CO], 31.8(2) [2×CH_3_(CH_2_)_5_*C*H_2_(CH_2_)_2_CO], 29.4, 29.3, 29.2, 29.1 [2×CH_3_(CH_2_)_3_(*C*H_2_)_2_(CH_2_)_3_CO], 27.0(2), 26.8, 26.7 [2×CH_3_CH_2_*C*H_2_CH_2_CO and 2×CH_3_(CH_2_)_2_*C*H_2_(CH_2_)_5_CO], 25.0, 24.9, 24.8, 24.7 [2×CH_3_CH_2_*C*H_2_*(*CH_2_)_4_CO and 2×CH_3_CH_2_*C*H_2_(CH_2_)_6_CO], 22.6, 22.2, 22.1(2) [2×CH_3_*C*H_2_(CH_2_)_2_CO and 2×CH_3_*C*H_2_(CH_2_)_7_CO], 14.0(2), and 13.6(2) [2×*C*H_3_(CH_2_)_3_CO and 2×*C*H_3_(CH_2_)_8_CO]. Anal. Calcd. for C_37_H_66_O_10_ (670.91): C, 66.24; H, 9.92. Found: C, 66.31; H, 9.88.

### 3.3. Predication of Biological Activities

The PASS (prediction of activity spectra for substances; http://www.way2drug.com/passonline/; accessed on 2 February 2021) concept was used for the analysis of the antimicrobial spectra of the synthesized compounds [41,42]. This web-based program can predict a plethora of useful biological activities including drug and non-drug actions, with a higher degree (90%) of accuracy. In general, PASS anticipated results are expressed as Pa (probability for active compound) and Pi (probability for inactive compound). In the present study, only Pa > Pi is considered for the biological activities of a compound on a scale of 0 to 1. On the other hand, the antiviral potentiality of the synthesized compounds was predicted using online software (http://crdd.osdd.net/servers/avcpred; accessed on 4 February 2021) and the results are expressed as a percentage of inhibition [53].

### 3.4. In Vitro Antimicrobial Potentiality Evaluation

Three pathogenic bacteria were selected for antibacterial evaluation. These are Gram-positive *Staphylococcus aureus* ATCC 25923, Gram-negative *Escherichia coli* ATCC 25922, and *Pseudomonas aeruginosa* CRL, ICDDR,B. Initially, a 2% solution of each compound was prepared in DMF (dimethylformamide). The diameter of the zone of inhibition against these bacterial organisms was determined following the disc diffusion method [40]. The results are compared with the standard drug tetracycline (Square Pharmaceuticals Ltd., Bangladesh). For more accuracy, each assay was conducted thrice.

Recently, infections caused by *Aspergillus* increased in many tropical and subtropical countries. Hence, the in vitro antifungal susceptibility was conducted against two human pathogenic fungi, viz., *Aspergillus fumigatus* ATCC 46645 and *Aspergillus niger* ATCC 16404. The activity was assessed by performing the food poisoning technique [54,55]. In this technique, the results are expressed in the percentage of linear mycelial growth inhibition and were measured after 2–4 days. For validation, the activity of fluconazole (Beximco Pharmaceuticals Ltd., Dhaka, Bangladesh) was measured under identical conditions.

### 3.5. Computational Methods

#### 3.5.1. Molecular Docking

As the synthesized MDG compounds **4**–**12** showed in vitro antimicrobial activity, especially against fungal pathogens, docking of the ligand-binding site(s) using in silico tools was attempted. First, we conducted virtual screening with all these structures against all the building blocks of the fungus cell membrane. Our derivatives showed better inhibition against sterol 14α-demethylase. Cytochrome P450 sterol 14α-demethylase is an important enzyme in ergosterol biosynthesis; ergosterol synthesis has been prevented by the inhibition of this enzyme, which further promotes cell membrane rupture in microorganisms. Fluconazole is currently the drug of choice, although the efficacy of the treatment is poor. For this purpose, to identify selective and potential inhibitors, we docked these synthetic compounds with cytochrome P450 sterol 14α-demethylase. Further, we analyzed their binding affinity and compared the docking score and the non-bond interaction with the reference VNI, standard fluconazole, and tetracycline.

The novel coronavirus (COVID-19), also known as SARS-CoV-2, is an enveloped RNA virus that has been declared a global pandemic by the World Health Organization. There is thus an urgent need for successful treatments and vaccines to be developed against this disease. In this context, we conducted a virtual screening of our derivatives against the main protease (M^pro^; PDB ID: 6LU7) of SARS-CoV-2. The structural similarity of our ligands with some of the medications used by different medical experts inspired us to test this hypothesis.

**Protein preparation.** Since cytochrome P450 sterol 14α-demethylase of fungus and the main protease (M^pro^) protein of SARS-CoV-2 were the molecular docking protein targets, their crystal structure was retrieved from the RCSB protein data bank [56] (PDB ID: 4UYL, organism: *Aspergillus fumigatus*; and PDB ID: 6LU7, organism: SARS-CoV-2). In parallel, both the 4UYL and 6LU7 protein crystal structures were prepared separately using the Maestro 11.6 software protein preparation wizard (Maestro, version 11.6, Schrödinger, LLC, New York, NY, USA), in which proper hydrogen, charges, and bond orders are initially assigned to the crystal structure. All the hydrogen bonds in the structure were optimized at a neutral pH, erasing unnecessary water. Then, the OPLS3e force field was applied in the minimization process, considering a structural deviation of not more than 0.30 Å of the RMSD. For molecular docking, the protein’s active site was fixed by creating a grid box at the reference ligand (VNI) binding site, in the case of 4UYL. For 6LU7, the active site was fixed by creating a grid box at the reference ligand inhibitor N3 (ID: 146025593) binding site. Grid generation parameters with a box size of 18Å × 18Å × 18Å and the OPLS3e force field are used post-minimization. The scaling factor of the charge cut-off and van der Waals were set at 0.25 and 1.00, respectively.

**Ligand preparation.** In order to construct a data set, after drawing in ChemDraw18.0, we collected the SDF format of these synthetic compounds. All the SDFs were prepared using Maestro 11.6 software’s LigPrep module 3.1 (Maestro, version 11.6, Schrödinger, LLC). Here, by implementing the OPLS3e force field, all the ligands were optimized. The module Epik 2.2 was used during the optimization to repair the ligands’ ionization state at pH 7.0 ± 2.0. Up to 32 possible stereoisomers of each compound were produced from this experiment and we selected the best low-energy conformer.

**Docking procedure.** Using the Glide module of the Maestro 11.6 program (Maestro, version 11.6, Schrödinger, LLC), we performed extra precision (XP) flexible docking, which is more advanced than SP/HTVS in the scoring feature. We conducted molecular docking separately for both the fungus enzyme and the main protease of SARS-CoV-2. Here, given the partial charge and the van der Waals factor of 0.15 and 0.80, respectively, all the ligands were conducted flexibly. After docking, minimization of the docked complex was performed using the OPLS3e force field. For each ligand, the best docked pose with the highest negative glide XP docking score value was recorded. We have analyzed and incorporated MM-GBSA binding affinity calculation of sterol 14α-demethylase and SEs complexes (Appendix A) and MM-GBSA binding affinity calculation of SARS-CoV-2 main protease (6LU7) and SEs complexes (Appendix A).

***In silico* inhibition constant (Ki) calculation**. A calculation of the efficacy of a substance in inhibiting a particular biological or biochemical activity is the half-maximum inhibitory concentration (IC_50_). This quantitative measure shows how much of a specific drug or other substance (inhibitor) is required to halve a given biological process. Hence, the theoretical IC_50_ was measured using AutoDock 4.2.2 [47]. We analyzed the inhibition constant (Ki) of six complexes, including the two reference ligand-protein complexes. Here, the inhibition constant (Ki) is directly proportional to the binding energy.

#### 3.5.2. Molecular Dynamics Simulations

The molecular dynamics simulation study was conducted in YASARA dynamics with the aid of the AMBER14 force field [57]. The complexes were cleaned and the hydrogen bond was optimized. The TIP3P water solvation model was used with periodic boundary conditions. The physiological conditions of the simulation system were set as 298 K, pH 7.4, and 0.9% NaCl. The initial energy minimizations were conducted by the steepest gradient approaches, with a simulated annealing method (5000 cycles). The time step of the simulation system was set as 2.0 fs. The long-range electrostatic interactions were calculated by the Particle Mesh Ewald method with a cut-off radius of 8.0Å. The simulation trajectories were saved after every 100 ps. Using the Berendsen thermostat, the simulations were conducted for 100 ns. The simulation trajectories were utilized to calculate the binding free energy by MM-PBSA methods in YASARA, where positive energy indicates more favorable bindings.

Binding Energy = EpotRecept + EsolvRecept + EpotLigand + EsolvLigand − EpotComplex − EsolvComplex [58].

#### 3.5.3. ADMET and Drug Friendliness Analysis

The pkCSM online tools [59] were used to predict the pharmacokinetic profile of the derivatives, where the absorption, distribution, metabolism, excretion, and toxicity (ADMET) of each ligand were measured. Using the SwissADME web tool [60], the drug-likeness and the medicinal chemical friendliness characteristics of the derivatives, such as the topological polar surface area (TPSA), Ghose filter, bioavailability score, and Lipinski’s rule of five were predicted. All the structures were first drawn by ChemDraw 18.0 software and the InChI key, SMILES, and SD file format were collected to use as input in various analyses. The InChI key has been used as a query in numerous databases to find these compounds and we have found that our derivatives are not yet mentioned in any database. In the pkCSM and SwissADME online tools, SMILES (simplified molecular-input line-entry system) strings were used, while the SD file format was used in molecular docking.

## 4. Conclusions

A one-step convenient method to synthesize a series of selective MDG-based novel SEs is presented. The applicability of these SEs as antimicrobials was tested. Both the PASS predication and in vitro evaluation established them as better antifungal agents than standard antibiotics such as fluconazole. The molecular docking score with sterol 14α-demethylase (4UYL), an important enzyme targeted by most of the antifungal agents, also supported this observation. Additionally, MDG esters **4**–**12** were found to inhibit the SARS-CoV-2 main protease, 6LU7. For future applications of all these promising results, ADMET and drug-likeness profiles were investigated, which indicated that these synthetic MDG esters are safe, non-toxic, and satisfied the necessary related tests. As the lack of effective antimicrobial and antiviral therapeutics constantly jeopardizes global public health, the study may be helpful for the development of environment-friendly biodegradable non-azole-type carbohydrate-based synthetic antimicrobials.

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
