# Peer review of "Novel Galactopyranoside Esters: Synthesis, Mechanism, In Vitro Antimicrobial Evaluation and Molecular Docking Studies"

_molecules, 2022, doi:10.3390/molecules27134125_

Round 1
Reviewer 1 Report
In this article, the authors reported the one-step direct unimolar valeroylation of methyl α-D-galactopyranoside to 6-O-valeroate along with DMAP catalyzed synthesis of 2,6-di-O-valeroate and 6-O-valeroate under a similar reaction condition. Further, the synthesized products are found to be biologically active which is quite promising. The current study is very interesting; however, the authors should address the following comments to improve the quality of the manuscript:
- The synthesis is very straightforward and provides little novelty in terms of the actual chemistry. However, the biological aspect is more interesting i. e antimicrobial and antiviral activity of synthesized compounds.
- The authors need to explain more clearly the logic of valeroylation than preparation of other sugar esters, was there some rationale to it?
- Authors did a fabulous job by showing chemical shifts, coupling constant values and pattern of multiplets for the newly synthesized compounds along with HSQC and HMBC. It will be nicer if authors can report specific rotations for all the newly synthesized compounds.
- it is recommended to cite following relevant article along with ref: 2 related to C-aryl glycosides which are stable glycosides:
Dubbu, S.; Chennaiah, A.; Verma, A. K.; Vankar, Y. D. Stereoselective synthesis of 2-deoxy-β-C-aryl/alkyl glycosides using Prins cyclization: Application in the synthesis of C-disaccharides and differently protected C-aryl glycosides. Carbohydr. Res. 2018, 468, 64−68.
Overall, after addressing the point mentioned above, I recommend this article to publish in molecules.
Author Response
[1] The synthesis is very straightforward and provides little novelty in terms of the actual chemistry. However, the biological aspect is more interesting i. e antimicrobial and antiviral activity of synthesized compounds.
Correction/modification: Thank you
[2] The authors need to explain more clearly the logic of valeroylation than preparation of other sugar esters, was there some rationale to it?
Correction/modification: We have checked that several esters of galactopyranosides are reported except valeroyl esters.
[3] Authors did a fabulous job by showing chemical shifts, coupling constant values and pattern of multiplets for the newly synthesized compounds along with HSQC and HMBC. It will be nicer if authors can report specific rotations for all the newly synthesized compounds.
Correction/modification/rebuttal: Thank you. Yes, it could be better. Unfortunately, we do not have such a facility right now. Definitely, we shall try in the near future.
[4] it is recommended to cite following relevant article along with ref: 2 related to C-aryl glycosides which are stable glycosides:
Correction/modification: Thank you. It is incorporated (new ref 2(b); highlighted in the manuscript).
Reviewer 2 Report
The manuscript needs substantial revision and additional assays to appreciate the quality.
It makes me very uncomfortable to see a study that has performed complex techniques such as synthesis and NMR analysis that fails so badly in the biological assays part.
Furthermore, the use of ADMET is unreliable. There are many intrinsic errors in the algorithms of all available servers. Although the proposal of pharmacokinetic prediction is something encouraging and very interesting, there are numerous inconsistencies between the data provided by ADMET servers and experimental data already available in the literature, as an example, a certain molecule is reported by ADMET server as not being substrate or inhibitor of some CYP enzyme, but experimental data already report that it is inhibitor, or that there is no some toxicity when experimental data already report that there is toxicity even at small concentrations, among others.
The authors themselves could see this in their experimental study: that molecules were mostly classified by the ADMET algorithm as no inhibitors of CYP P450, so, why do a docking study with this target? The azole fungicides (positive control) act in the fungal cell by preventing the conversion of lanosterol to ergosterol, a conversion that is prevented by inhibition of an enzyme of the P450 complex.
Furthermore, many more experimental studies (in vitro and in vivo) evaluating the activity of these compounds - including their toxicity - are needed before these synthetic derivatives can be effectively considered as an alternative for combating these microorganisms
The result of the data with the SARS-CoV-2 Mpro has not the slightest logic to be in this manuscript.
There are no MD simulation studies or binding calculation studies are performed, there is no energy calculations by MMPBSA/MMGBSA or similar approaches. Without MD simulation, just on the basis of docking the prediction has limited scope and applications. Thus, it is essential to carryout the MD simulation studies.
Author Response
[1] The manuscript needs substantial revision and additional assays to appreciate the quality.
Correction/modification: Thank you. We have improved/modified several sections as per all the reviewers’ comments.
[2] It makes me very uncomfortable to see a study that has performed complex techniques such as synthesis and NMR analysis that fails so badly in the biological assays part.
Correction/modification: --
[3] Furthermore, the use of ADMET is unreliable. There are many intrinsic errors in the algorithms of all available servers. Although the proposal of pharmacokinetic prediction is something encouraging and very interesting, there are numerous inconsistencies between the data provided by ADMET servers and experimental data already available in the literature, as an example, a certain molecule is reported by ADMET server as not being substrate or inhibitor of some CYP enzyme, but experimental data already report that it is inhibitor, or that there is no some toxicity when experimental data already report that there is toxicity even at small concentrations, among others. The authors themselves could see this in their experimental study: that molecules were mostly classified by the ADMET algorithm as no inhibitors of CYP P450, so, why do a docking study with this target? The azole fungicides (positive control) act in the fungal cell by preventing the conversion of lanosterol to ergosterol, a conversion that is prevented by inhibition of an enzyme of the P450 complex. Furthermore, many more experimental studies (in vitro and in vivo) evaluating the activity of these compounds - including their toxicity - are needed before these synthetic derivatives can be effectively considered as an alternative for combating these microorganisms.
Correction/modification/rebuttal: Thank you. We will try the honorable reviewer’s suggestion in the near future.
[4] The result of the data with the SARS-CoV-2 Mpro has not the slightest logic to be in this manuscript. There are no MD simulation studies or binding calculation studies are performed, there is no energy calculations by MMPBSA/MMGBSA or similar approaches. Without MD simulation, just on the basis of docking the prediction has limited scope and applications. Thus, it is essential to carryout the MD simulation studies.
Correction/modification: Thank you. The molecules showed better antifungal efficacy. Therefore, we conducted molecular docking with antifungal protein sterol 14α-demethylase (PDB ID: 4UYL). As many antifungals are recently investigated for the probable SARS-CoV-2 main protease inhibitors, we have also checked the similar probability of these synthesized compounds.
We have conducted 100 ns molecular dynamics simulations to understand the stability of the docked complexes and duly in the manuscript (highlighted yellow in the manuscript). The MM-PBSA binding free energy was also conducted, and over simulations, the study demonstrates the stable nature of the complexes. MM-GBSA binding affinity calculation of (i) sterol 14α-demethylase and SEs complexes and (ii) SARS-CoV-2 main protease (6LU7) and SEs complexes are added in the supplementary file.
Reviewer 3 Report
The work on the “ Novel Galactopyranoside Esters: Synthesis, Mechanism, In Vitro Antimicrobial Evaluation, and Molecular Docking Studies” is a valuable well-done contribution.
An extensive English revision is needed and typo errors were observed in the whole manuscript.
- I would like to recommend the authors to re-write a lot of sentences due to the overlapping and high similarity rate with a lot of other articles especially the last works for the same authors “10.1016/j.carres.2020.108130”, the sentences were copied as its, and the similarity rate was too high. The following lines should be edited accordingly: 42-45, 53-56, 81-84, 225-231, 234-238, 347-350, 351-360, 362-410, 490-500, and 612-645
The in-vitro and the bacterial and fungal strains should be written in italic in the whole manuscript.
Figures 5 and 6 need improvement because some of the amino acids are not clear.
A high percentage of self-citations for the corresponding author Mohammed M. Matin,
According to the final compound, characterizations did the authors used HRMS or elemental analysis to confirm the structures ??
Author Response
[1] The work on the “ Novel Galactopyranoside Esters: Synthesis, Mechanism, In Vitro Antimicrobial Evaluation, and Molecular Docking Studies” is a valuable well-done contribution.
Correction/modification: Thank you.
[2] An extensive English revision is needed and typo errors were observed in the whole manuscript.
Correction/modification/rebuttal: Thank you. We have checked the full manuscript and improved it accordingly (highlighted in the manuscript).
[3] I would like to recommend the authors to re-write a lot of sentences due to the overlapping and high similarity rate with a lot of other articles especially the last works for the same authors “10.1016/j.carres.2020.108130”, the sentences were copied as its, and the similarity rate was too high. The following lines should be edited accordingly: 42-45, 53-56, 81-84, 225-231, 234-238, 347-350, 351-360, 362-410, 490-500, and 612-645.
Correction/modification: Thank you. Modified the manuscript accordingly (highlighted yellow).
[4] The in-vitro and the bacterial and fungal strains should be written in italic in the whole manuscript.
Correction/modification: Thank you. Modified accordingly.
[5] Figures 5 and 6 need improvement because some of the amino acids are not clear.
Correction/modification: Thank you. Modified accordingly.
[6] A high percentage of self-citations for the corresponding author Mohammed M. Matin.
Correction/modification: Thank you. We believe that these references are necessary and related to the subjects described.
[7] According to the final compound, characterizations did the authors used HRMS or elemental analysis to confirm the structures??
Correction/modification/rebuttal: Thank you. Yes, it could be better. Unfortunately, we do not have an HRMS facility right now. Definitely, we shall try in the near future.
Reviewer 4 Report
Matin, et al, presented the synthesis of novel galactopyranoside esters. Also the in vitro antimicrobial evaluation and molecular docking are studied. Their results suggest that the synthesized compounds had better anti-fungal activities over antimicrobial evaluation. To rationalize the results molecular docking was conducted and it supports the in vitro antifungal activities. Although the synthesis of the galactopyranoside esters seems to be accessible the compounds could be helpful for the development of synthetic antimicrobials and antivirals compounds.
In general, the manuscript may be fascinating to the scientific community. I recommend being accepted after minor changes are made.
1. Line 30: Use italic in "in vitro"
2. Line 37: Change antimicrobial by antifungal
The authors describe the Elemental Analysis for compounds 4, 5, 9, 10, 11, and 12. It is suggested to get the High-Resolution Mass Spectrometry. For instance HR-MS-QTOF.
The terms downfield in lines 114, 139, 176, and 196 are obsolete. Nowadays is preferred to use high or low frequencies.
Author Response
[1] Matin, et al, presented the synthesis of novel galactopyranoside esters. Also the in vitro antimicrobial evaluation and molecular docking are studied. Their results suggest that the synthesized compounds had better anti-fungal activities over antimicrobial evaluation. To rationalize the results molecular docking was conducted and it supports the in vitro antifungal activities. Although the synthesis of the galactopyranoside esters seems to be accessible the compounds could be helpful for the development of synthetic antimicrobials and antivirals compounds. In general, the manuscript may be fascinating to the scientific community. I recommend being accepted after minor changes are made.
Correction/modification: Thank you.
[2] 1. Line 30: Use italic in "in vitro"
Correction/modification: Thank you. Modified accordingly (highlighted in the manuscript).
[3] 2. Line 37: Change antimicrobial by antifungal
Correction/modification: Thank you. Modified accordingly.
[4] The authors describe the Elemental Analysis for compounds 4, 5, 9, 10, 11, and 12. It is suggested to get the High-Resolution Mass Spectrometry. For instance HR-MS-QTOF.
Correction/modification/rebuttal: Thank you. Yes, it could be better. Unfortunately, we do not have such a facility right now. Definitely, we shall try in the near future.
[5] The terms downfield in lines 114, 139, 176, and 196 are obsolete. Nowadays is preferred to use high or low frequencies.
Correction/modification: Thank you. We have modified the manuscript accordingly.
Round 2
Reviewer 2 Report
While the manuscript has indeed improved in many respects, the experimental part still needs A LOT - a lot really - of analysis.
The experimental trials were the most basic and simple in existence, and give little information about the mechanism of action of the proposed compounds.
The authors who by force, try convince that because the in silico study reports that there is interaction between the compounds and the CYP structure and that because they have shown a certain activity in the basic tests, that this activity is probably by inhibition of CYP. With the data presented by the study, it is very presumptuous and risky to say that.
Besides, the truth is that the study lacks focus. They want to shoot at all sides and hit anything: bacteria, fungi and viruses.
Why was there no computational study with bacteria, only with fungi?
Why hasn't there been any kind of experimental study with SARS-CoV-2? There are already many kits for in vitro assay that only need a plate reader, without presenting risks to the users...
The study needs to focus and really define what it wants to address and perform the other minimum tests pertinent to the study of the type of microorganism that is proposed.
Author Response
We are thankful to the reviewer 2 for his valuable comments. In this respect, corrections and our response are as follows:
Comment [1]. While the manuscript has indeed improved in many respects, the experimental part still needs A LOT - a lot really - of analysis.
- Thank you for your valuable comment. We humbly inform you that due to our insufficient funds we are unable go further.
Comment [2]. The experimental trials were the most basic and simple in existence, and give little information about the mechanism of action of the proposed compounds.
- Thank you for your valuable comment. Our principal aim was one-step synthesis and characterization of galactopyranoside esters, which we have achieved successfully. To show the applicability of synthesized compounds in vitro antimicrobial tests were conducted in the laboratory. In favor/support of the obtained antimicrobial results, we conducted/extended molecular docking, dynamics simulation (also suggested by one honorable reviewer), etc.
Comment [3]. The authors who by force, try convince that because the in silico study reports that there is interaction between the compounds and the CYP structure and that because they have shown a certain activity in the basic tests, that this activity is probably by inhibition of CYP. With the data presented by the study, it is very presumptuous and risky to say that.
- Thank you for your valuable comment. In this respect, our analysis/views are as follows (also we have added it in the new revised manuscript; highlighted in yellow).
The molecular docking results by targeting CYP and ligand molecules indicates that multiple interactions were observed at the active sites of sterol 14α-demethylase; LEU125, LEU503, TYR122, PHE229, ALA307, ILE373, LEU503, LEU125, TYR136, LYS147, ILE464, LEU367, CYS464, ILE376, LYS147, PHE234, and PHE504 residues. The multiple interactions at the active site cavity by the ligand molecules may interfere the functions of the targeted protein or enzymes [31]. Therefore, multiple trajectories from molecular dynamics simulations study demonstrate the rigid and stable nature of the docked complexes. Based on the findings it may be concluded that the ligand molecules may inhibit the CYP but this data need to be validated from wet lab experiments.
[31] Srinivasan, S., Sadasivam, S. K., Gunalan, S., Shanmugam, G., & Kothandan, G. (2019). Application of docking and active site analysis for enzyme linked biodegradation of textile dyes. Environmental Pollution, 248, 599-608.
Comment [4]. Besides, the truth is that the study lacks focus. They want to shoot at all sides and hit anything: bacteria, fungi and viruses.
- Thank you for your valuable comment. In the introduction section (2nd paragraph), we have mentioned that galactopyranose salicylic esters showed notable antimicrobial efficacy [Ref 17] like many other sugar esters. Rationally, the synthesized compounds were checked for their antimicrobial activities in vitro. To check/explore the new activity/activities we have checked their antiviral efficacy also. The obtained results need to be validated in future from wet lab experiments.
Comment [5]. Why was there no computational study with bacteria, only with fungi?
- Thank you for your valuable comment. Based on the better PASS calculation (Table 1) and in vitro antifungal results (Table 4) than antibacterial results (Table 1 and 3), we have conducted molecular docking with the antifungal enzyme (4UYL). This reason is already mentioned in section 2.7 (reviewer may kindly check, highlighted in yellow).
Comment [6]. Why hasn't there been any kind of experimental study with SARS-CoV-2? There are already many kits for in vitro assay that only need a plate reader, without presenting risks to the users...
- We thank the reviewer for his suggestion. Please be informed that due to lower funding, we are unable to conduct the experimental study against SARS-CoV-2. However, we will try to conduct the experimental validations for the next projects in future.
Comment [7]. The study needs to focus and really define what it wants to address and perform the other minimum tests pertinent to the study of the type of microorganism that is proposed.
- Thank you for your valuable comment. We humbly inform you that due to our insufficient funds we are unable go further. In addition, our principal objective was successfully achieved (Response and comment number 2,5,6).
Reviewer 3 Report
the manuscript was improved
Best Regards
Author Response
Thank you for your kind review and time.